# CAT: COLLABORATIVE ADVERSARIAL TRAINING

## ABSTRACT

Adversarial training has proven to be effective in enhancing the robustness of neural networks. However, previous methods typically focus on a single adversarial training strategy and do not consider the characteristics of models trained by different strategies. Upon revisiting these methods, we have observed that different adversarial training methods exhibit distinct levels of robustness for sample instances. For instance, a model trained by AT may correctly classify a sample instance that is misclassified by a model trained by TRADES, and vice versa. Motivated by this observation, we propose a Collaborative Adversarial Training (**CAT**) framework to enhance the robustness of neural networks. CAT utilizes different adversarial training methods to train robust models and facilitate the interaction of these models to leverage their combined knowledge during the training process. Extensive experiments conducted on various networks and datasets validate the effectiveness of our method.

## 1 INTRODUCTION

With the advancements in deep learning, Deep Neural Networks (DNNs) have been widely applied to various visual tasks, such as image classification (He et al., 2016), object detection (Redmon et al., 2016), and semantic segmentation (Pal & Pal, 1993). These networks have achieved state-of-the-art performance. However, recent research has revealed that DNNs are vulnerable to adversarial perturbations (Goodfellow et al., 2014). A carefully crafted adversarial perturbation by a malicious agent can easily deceive the neural network, which raises security concerns, particularly in security-critical areas like autonomous driving (Chen et al., 2019). To address the vulnerability of DNNs, various methods have been proposed, including adversarial training (Madry et al., 2017), defensive distillation (Papernot et al., 2016), feature denoising (Xie et al., 2019), and neural network pruning (Madaan et al., 2020). Among them, Adversarial Training (AT) is considered the most effective method for improving adversarial robustness. AT can be viewed as a data augmentation strategy that trains neural networks using adversarial examples crafted from natural examples. Typically, AT is formulated as a min-maximization problem, where the inner maximization generates adversarial examples, and the outer minimization optimizes the model's parameters based on the adversarial examples generated during the inner maximization process. However, previous approaches have primarily focused on enhancing a model's adversarial accuracy, without considering the distinctive characteristics of different methods. This prompts us to question whether models trained by different adversarial training methods perform similarly on individual instances.

In the analysis of various adversarial training methods, we discovered classification inconsistencies among models trained by different techniques, as illustrated in Fig. 1. For instance, when considering AT (Madry et al., 2017) and TRADES (Zhang et al., 2019), the network trained with AT may correctly classify a given adversarial example, while the network trained with TRADES misclassifies it, and vice versa. Consequently, we can deduce that although AT and TRADES exhibit similar numerical adversarial accuracy, they behave differently on individual instances, implying that models trained by different methods acquire diverse knowledge. This leads us to question:

*Do two networks learn better if they collaborate?*

Motivated by this observation, we propose the **C**ollaborative **A**dversarial **T**raining (**CAT**) framework to enhance the robustness of neural networks. As illustrated in Fig. 2, our framework simultaneously trains two separate deep neural networks using different adversarial training methods. Specifically, adversarial examples generated by one network are utilized as input for the other network, enabling

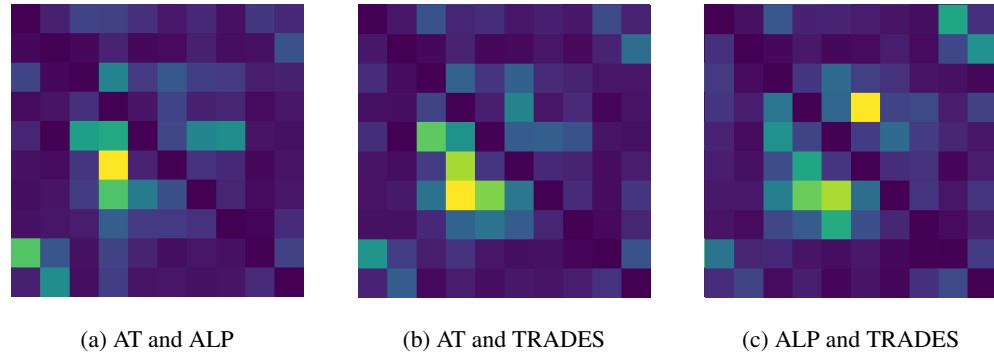

| (a) AT and ALP | (b) AT and TRADES | (c) ALP and TRADES |

Figure 1: **Confusion matrices of models trained by different methods** with ResNet-18 on the CIFAR-10 test dataset (better seen with color). We set the diagonal value as 0 for better illustration. Confusion exists in models trained by any two methods, especially for blocks from class 3 to class 7. The value of the prediction discrepancy is 18.98%, 22.54%, and 21.05% respectively.

the exchange of knowledge to guide their respective learning processes. By facilitating collaborative learning, we improve the overall robustness of the neural networks. Extensive experiments conducted on various datasets (CIFAR, Tiny-ImageNet) and neural networks (VGG, MobileNet, ResNet) demonstrate the effectiveness of our approach. Under the Auto-Attack benchmark, CAT achieves state-of-the-art robustness on CIFAR-10 without the need for additional synthetic or real data. Furthermore, we provide a comprehensive property analysis of CAT to deepen our understanding of its mechanisms. In summary, our contributions are threefold:

- We observe that models trained by different adversarial training methods exhibit distinct characteristics for individual sample instances.
- We introduce Collaborative Adversarial Training (CAT), a novel framework that trains neural networks simultaneously using different adversarial training methods.
- Extensive experiments on diverse datasets and networks demonstrate the effectiveness of CAT. CAT achieves new state-of-the-art performance without the need for additional data.

## 2 RELATED WORK

### 2.1 ADVERSARIAL ATTACK

**White-box Attack:** Goodfellow (Goodfellow et al., 2014) proposes FGSM to efficiently craft adversarial examples, which can be generated in just one step. Madry proposes PGD to generate adversarial examples, which is the most efficient way of using the first-order information of the network. MI-FGSM (Dong et al., 2018) combines momentum into the iterative process to help the model escape from local optimal points. And the adversarial examples generated by this method are also more transferable. Boundary-based attacks such as deepfool (Moosavi-Dezfooli et al., 2016) and CW (Carlini & Wagner, 2017) also make the model more challenging. Recently, the ensemble approach of diverse attack methods (Auto-Attack), consisting of APGD-CE (Croce & Hein, 2020b), APGD-DLR (Croce & Hein, 2020b), FAB (Croce & Hein, 2020a) and Square Attack (Andriushchenko et al., 2020), become a benchmark for testing model robustness.

**Black-box Attack:** Block-box attacks can be categorized into transfer-based and query-based attacks. Transfer-based methods attack the target model by using the transferability of adversarial examples, *i.e.*, the adversarial examples generated on the surrogate model can be transferred to fool the target model. There are many ways to explore the transferability of adversarial examples for black-box attacks. Dong (Dong et al., 2018) combines momentum with an iterative approach to obtain better transferability. Scale-invariance (Lin et al., 2019) boosts the transferability of adversarial examples by transforming the inputs on multiple scales. Square Attack (Andriushchenko et al., 2020) approximates model's decision boundary based on a randomized search scheme to be the most efficient query-based attack method.

## 2.2 ADVERSARIAL ROBUSTNESS

Adversarial attacks present a significant threat to DNNs. For this reason, many methods have been proposed to defend against adversarial examples, including denoising (Xie et al., 2019), adversarial training (Madry et al., 2017), data aumentation (Rebuffi et al., 2021), and input purification (Naseer et al., 2020). ANP (Madaan et al., 2020) finds the vulnerability of latent features and uses pruning to improve robustness. Madry uses PGD to generate adversarial examples for adversarial training, which is also the most effective way to defend against adversarial examples. A large body of work uses new regularization or objective functions to improve the effectiveness of standard adversarial training. Adversarial logit pairing (Kannan et al., 2018) improves robustness by encouraging the logits of normal and adversarial examples to be closer together. TRADES (Zhang et al., 2019) uses KL divergence to regularize the output of adversarial and pure examples.

## 2.3 KNOWLEDGE DISTILLATION

Knowledge distillation (KD) is commonly used for model compression and was first used by Hinton (Hinton et al., 2015) to distill knowledge from a well-trained teacher network to a student network. KD can significantly improve the accuracy of student models. There have been many later works to improve the effectiveness of KD (Romero et al., 2014). In recent years, KD has been extended to other areas. Goldblum (Goldblum et al., 2020) analyzes the application of knowledge distillation to adversarial robustness and proposes ARD to transfer knowledge from a large teacher model with better robustness to a small student model. ARD can produce a student network with better robustness than training from scratch. In this paper, we propose a more effective collaborative training framework to improve the robustness of the network.

## 3 PROPOSED METHOD

In this section, we use the methods of Adversarial Training (AT) and TRADES as examples to introduce Collaborative Adversarial Training (CAT). We provide a brief overview of the training objective functions of AT and TRADES, followed by a detailed introduction of CAT.

## 3.1 PRELIMINARY

Adversarial training can be formulated as a min-maximization problem. It utilizes Projected Gradient Descent (PGD) to generate adversarial examples for the internal maximization process, while the external minimization optimizes the model parameters using the PGD-generated adversarial examples and the ground-truth label $y$. The objective function of AT is defined as:

$$\min_{\theta} \mathbb{E}_{(x,y)\in D_{data}}(\arg\max_{\delta} L(f_\theta(x_{AT}^{adv}), y)), \tag{1}$$

$$x_{AT}^{adv} = x + \delta, \tag{2}$$

where $D_{data}$ represents the distribution of training data, $x$ and $y$ denote the training data and corresponding labels from $D_{data}$. $f_\theta$ represents a neural network parameterized by $\theta$. $L$ denotes the standard cross-entropy loss commonly used in image classification tasks. $\delta$ represents the adversarial perturbations generated by PGD.

Neural networks trained by AT can achieve a certain level of robustness, but this often comes at the cost of decreased accuracy on natural examples. To address this problem, TRADES constraints the distribution distance of adversarial and natural features, introducing a new training objective function. It is formulated as follows:

$$\min_{\theta'} \mathbb{E}_{(x,y)\in D_{data}} L(g_{\theta'}(x), y) + \lambda D_{KL}(g_{\theta'}(x), g_{\theta'}(x_{TRADES}^{adv})), \tag{3}$$

where, $x_{TRADES}^{adv}$ represents the adversarial data corresponding to the natural data $x$, and $D_{KL}$ represents the Kullback-Leibler (KL) divergence used to align the natural logits and adversarial logits. The trade-off between the natural accuracy and distribution distance is balanced by the parameter $\lambda$.

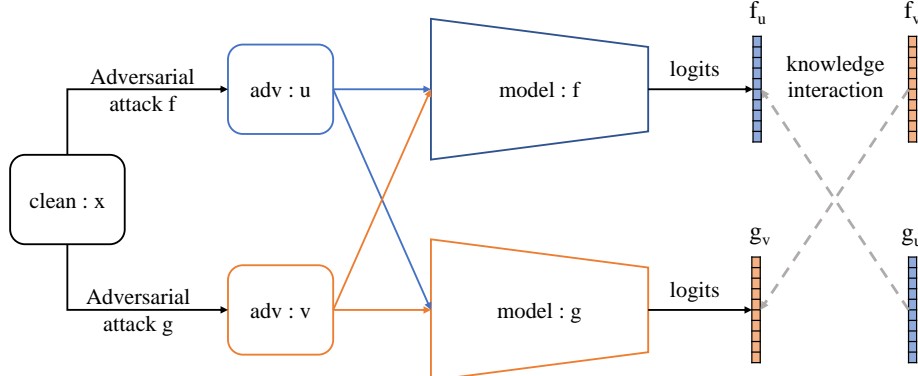

Figure 2: **The framework of CAT, performing adversarial training collaboratively.** Given a batch of natural examples, the two networks $f$ and $g$ are attacked separately to generate adversarial examples u and v. Then u and v are fed into both networks to obtain the corresponding logits. We then use the logits obtained from the peer networks to guide the learning of its network, *i.e.*, $g_u \to f_u, f_v \to g_v$. The process is called knowledge interaction.

## 3.2 Collaborative Adversarial Training

Motivated by the observation that models trained by different methods acquire distinct knowledge. Collaborative Adversarial Training (CAT) aims to enhance robustness by facilitating knowledge sharing and interaction among neural networks trained by different methods through collaborative learning. The framework is illustrated in Fig. 2. CAT utilizes the knowledge of a peer network, which is trained by a different method, to guide the learning process of a given network.

Specifically, the adversarial data generated by the network trained by Adversarial Training (AT) are fed into the peer network trained by TRADES, obtaining the corresponding logit. This logit is then used to guide the network training by AT, which is formulated as:

$$L_1 = D_{KL}(f(x_f^{adv}), \hat{g}(x_f^{adv})), \tag{4}$$

where, $f$ represents the network trained by AT, $g$ represents the network trained by TRADES, and $\hat{g}(x_f^{adv})$ denotes the logit obtained from the network trained by TRADES. $\hat{}$ denotes that we take the logit as a constant. $x_f^{adv}$ represents the adversarial data generated by $f$ using PGD.

Similarly, to enable collaborative learning, we feed the adversarial examples generated by the TRADES network to the AT network to obtain the corresponding logit. This logit is used to guide the network training by TRADES. The loss can be formulated as:

$$L_2 = D_{KL}(g(x_g^{adv}), \hat{f}(x_g^{adv})). \tag{5}$$

Experimentally, models trained solely using collaborative loss tend to collapse due to the lack of supervision for guiding the knowledge-sharing process. To address this problem, we introduce supervision by combining the respective training objective functions of the two networks with the collaborative learning objective function. Therefore, the training objective function for collaborative adversarial training based on AT and TRADES can be formulated as:

$$L_{total} = \alpha L_{TRADES} + (1 - \alpha)L_2 + \alpha L_{AT} + (1 - \alpha)L_1, \tag{6}$$

where $\alpha$ is a trade-off parameter balancing the guidance of peer network knowledge and the original objective function. $L_{TRADES}$ represents the training objective of TRADES as defined in Eq. (3), and $L_{AT}$ represents the training objective of AT as defined in Eq. (1). The first two terms in Eq. (6) are used to train model $g$, while the last two terms are used to train model $f$.

The decision boundaries learned by different adversarial training methods can vary. However, under the guidance of peer network knowledge, as described in Eq. (4) and Eq. (5), the two networks trained by different methods continuously optimize the classification decision boundaries through

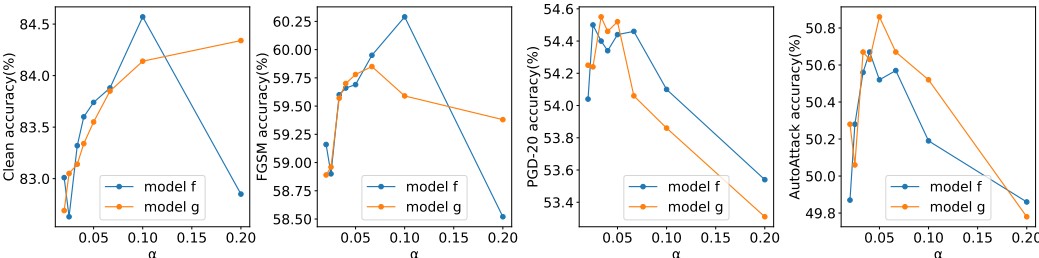

Figure 3: **Adversarial robustness using different hyperparameters of CAT.** From left to right, the results of Clean acc, FGSM acc, PGD acc, and AA acc are shown. model $f$ and model $g$ represent the results of using TRADES and AT in the CAT training framework, respectively.

collaborative learning. As a result, both networks learn improved decision boundaries compared to learning in isolation, leading to enhanced adversarial robustness.

Collaborative Adversarial Training (CAT) is a generalized adversarial training method that can be applied to any two adversarial methods. Moreover, CAT can incorporate any number of different adversarial training methods for collaborative learning. The results of using CAT with three adversarial training methods are reported in Sec. 5.3.

**Difference from ensemble methods:** A similar work with our CAT is ensemble learning. The main distinction between collaborative adversarial learning and ensemble methods lies in the learning process. Collaborative learning involves multiple models that learn from each other, while ensemble methods involve multiple models that are combined to produce a single output. During testing, each model trained through collaborative learning makes individual predictions, as they have interacted with each other's knowledge during the training phase. Ensemble methods involve multiple models all the time. For more details, please refer to Appendix B.

## 4 EXPERIMENTS

In this section, we conduct extensive experiments on popular benchmark datasets to demonstrate the effectiveness of CAT. First, we briefly introduce the experiment setup and implementation details. Then, ablation studies are done to choose the best hyperparameters and CAT methods. Finally, according to the best CAT methods, we report the white-box and black-box adversarial robustness on two popular benchmark datasets.

**Training setup:** Our overall training parameters refer to Madaan et al. (2020). Specifically, we use SGD (momentum 0.9, batch size 128) to train ResNet18 for 200 epochs on the CIFAR-10 dataset with weight decay 5e-4 and initial learning rate 0.1 which is divided by 10 at 100-th and 150-th epoch, respectively. For the internal maximization process, we use $PGD_{10}$ adversarial attack to solve, with a random start, step size 2.0/255, and perturbation size 8.0/255.

**Evaluation setup:** We report the clean accuracy on natural examples and the adversarial accuracy on adversarial examples. For adversarial accuracy, we report both white-box and black-box robustness, following the widely used protocols in the adversarial research field. For the white-box attack, we consider three basic attack methods: FGSM (Goodfellow et al., 2014), PGD (Madry et al., 2017), and $CW_{\infty}$ (Carlini & Wagner, 2017) optimized by $PGD_{20}$, and a stronger ensemble attack method named AutoAttack (AA) (Croce & Hein, 2020b). For the black-box attacks, we consider both transfer-based attacks and query-based attacks.

### 4.1 ABLATION STUDY

#### 4.1.1 HYPERPARAMETER

CAT improves adversarial robustness through the learning of collaboration, which requires both the knowledge of peer networks and the guidance of the ground truth label. The balance of these two items is traded off by a hyperparameter $\alpha$. We execute collaborative training by TRADES and AT

Table 1: **The white-box robustness of different CAT methods on CIFAR-10.** We report the results of the best checkpoint and last checkpoint. ResNet-18 is the basic network in CAT framework.

| Method | Best Checkpoint | | | | | Last Checkpoint | | | | |
|---|---|---|---|---|---|---|---|---|---|---|
| | Clean | FGSM | PGD$_{20}$ | CW$_\infty$ | AA | Clean | FGSM | PGD$_{20}$ | CW$_\infty$ | AA |
| CAT$_{AT-TRADES}$ | 83.74 | 59.69 | 54.44 | 52.60 | 50.52 | 84.45 | **60.03** | **53.01** | **52.01** | 49.30 |
| | 83.55 | 59.78 | **54.52** | 52.58 | 50.86 | 84.12 | 59.69 | 52.82 | 51.88 | 49.39 |
| CAT$_{AT-ALP}$ | 84.66 | 59.94 | 53.11 | 51.90 | 49.74 | 84.71 | 59.84 | 50.77 | 50.53 | 47.80 |
| | **85.21** | **60.21** | 53.02 | 52.13 | 49.96 | **85.27** | 59.75 | 51.10 | 50.69 | 47.91 |
| CAT$_{TRADES-ALP}$ | 83.91 | 59.76 | 54.44 | 52.56 | **51.02** | 84.67 | 59.85 | 52.51 | 51.43 | 49.31 |
| | 84.75 | 59.76 | 54.17 | **52.72** | 50.85 | **85.27** | 59.82 | 52.56 | 51.83 | **49.64** |

Table 2: **The white-box robustness of CAT on CIFAR-10 and CIFAR-100.** We report the results of the best checkpoint and last checkpoint. ResNet-18 is the basic network in CAT framework.

| Dataset | Method | Best Checkpoint | | | | | Last Checkpoint | | | | |
|---|---|---|---|---|---|---|---|---|---|---|---|
| | | Clean | FGSM | PGD$_{20}$ | CW$_\infty$ | AA | Clean | FGSM | PGD$_{20}$ | CW$_\infty$ | AA |
| CIFAR-10 | Natural | **94.65** | 19.26 | 0.0 | 0.0 | 0.0 | **94.65** | 19.26 | 0.0 | 0.0 | 0.0 |
| | AT | 82.82 | 57.57 | 51.76 | 50.05 | 47.55 | 84.53 | 53.90 | 43.56 | 44.19 | 41.57 |
| | TRADES | 83.17 | 59.22 | 52.63 | 50.79 | 49.21 | 83.04 | 57.46 | 49.81 | 49.01 | 47.03 |
| | ALP | 83.85 | 57.20 | 51.88 | 50.11 | 48.48 | 84.64 | 55.35 | 44.96 | 44.54 | 42.62 |
| | CAT | 83.91 | **59.76** | **54.44** | 52.56 | **51.02** | 84.67 | **59.85** | 52.51 | 51.43 | 49.31 |
| | | 84.75 | **59.76** | 54.17 | **52.72** | 50.85 | 85.27 | 59.82 | **52.56** | **51.83** | **49.64** |
| CIFAR-100 | Natural | **75.55** | 9.48 | 0.0 | 0.0 | 0.0 | **75.39** | 9.57 | 0.0 | 0.0 | 0.0 |
| | AT | 57.42 | 31.90 | 28.78 | 27.27 | 24.88 | 57.34 | 26.77 | 21.24 | 21.50 | 19.59 |
| | TRADES | 56.98 | 31.72 | 29.04 | 25.30 | 24.23 | 55.08 | 30.40 | 26.81 | 24.78 | 23.68 |
| | ALP | 61.01 | 31.41 | 26.78 | 25.68 | 23.51 | 58.4 | 27.97 | 22.63 | 21.87 | 20.42 |
| | CAT | 61.31 | 35.83 | **33.09** | **29.17** | **27.17** | 61.78 | **35.84** | **32.76** | **29.48** | **27.29** |
| | | 62.53 | **36.05** | 32.92 | 29.16 | 26.90 | 62.52 | 35.79 | 32.51 | 29.24 | 26.73 |

as the base method and experiment with different values of $\alpha$. The experiment results are illustrated in Fig. 3. From the figure, we can conclude that if $\alpha$ is too high, *i.e.*, little knowledge is extracted from the peer network, the effect is about the same as training with AT and trades alone. If $\alpha$ is too small, *i.e.*, overly focused on the knowledge obtained from the peer network, The network is vulnerable and will collapse when $\alpha = 0$, which is not shown. Since Auto-Attack is currently the most powerful integrated attack method, we choose hyperparameters $\alpha$ based primarily on the robustness of the network against AA. In the following experiments, $\alpha$ is set for 0.05 by default.

### 4.1.2 Different CAT methods

As described in Sec. 3.2, any two adversarial training methods can be incorporated into the CAT framework and learned collaboratively. Considering that different adversarial training methods have distinct properties, the performance of different CAT methods may also vary. For this reason, we consider three collaborative adversarial training methods, *i.e.*, AT-TRADES, AT-ALP, and TRADES-ALP. Due to the fact that CAT uses two models for collaborative training, we report the results for both networks. Tab. 1 shows the performance of CAT using different adversarial training methods. CAT achieves good robustness against four attack methods in all settings. We again mainly consider the performance of AA and choose TRADES-ALP as the base method for CAT. Without a further statement, CAT represents CAT$_{TRADES-ALP}$ in the following sections. Further, we analyze the correlation between discrepancy and performance after collaborative learning, which is delayed to Sec. 5.2. The results of CAT in other settings are delayed to Appendices A.4 and A.5

Table 3: **The black-box robustness of CAT on CIFAR-10 and CIFAR-100.** We only report the results of the best checkpoint. ResNet-18 is the basic network in CAT framework.

| Method | CIFAR-10 | | | | | CIFAR-100 | | | | |
|---|---|---|---|---|---|---|---|---|---|---|
| | FGSM | $PGD_{20}$ | $PGD_{40}$ | $CW_\infty$ | Square | FGSM | $PGD_{20}$ | $PGD_{40}$ | $CW_\infty$ | Square |
| AT | 64.54 | 61.70 | 61.57 | 61.42 | 56.16 | 39.15 | 37.56 | 37.46 | 38.85 | 30.11 |
| TRADES | 65.63 | 63.57 | 63.57 | 63.23 | 55.97 | 39.06 | 37.73 | 37.79 | 38.86 | 28.72 |
| ALP | 64.95 | 62.38 | 62.32 | 61.78 | 55.78 | 40.29 | 38.97 | 38.85 | 40.03 | 29.85 |
| CAT | 65.73 | 63.65 | 63.78 | 63.24 | 57.55 | 42.26 | 40.76 | 40.76 | 41.78 | 33.04 |
| | **66.06** | **63.91** | **63.88** | **63.26** | **57.95** | **42.81** | **41.55** | **41.42** | **42.42** | **33.30** |

## 4.2 ADVERSARIAL ROBUSTNESS

### 4.2.1 WHITE-BOX ROBUSTNESS

For FGSM, PGD, $CW_\infty$, AA, the attack perturbations are all 8.0/255 and the step size for PGD, $CW_\infty$ are 2/25, with 20 iterations. We report the results of both the best and the last checkpoint. The best checkpoint result of the training phase is selected based on the model's PGD defense for the test dataset (attack step size 2.0/255, iteration number 10, perturbation size 8.0/255).

Tab. 2 shows the adversarial accuracy of the networks trained by different methods on CIFAR-10 and CIFAR-100 against the four attacks. We also report the accuracy of the model for natural examples. From the table, we can obtain the following conclusions: (1) CAT obtains good robustness against all four attacks on both datasets. For example, for the strongest AA attack method, CAT can obtain 2% improvement. (2) CAT obtains high adversarial robustness while ensuring accuracy for natural examples. Although there is still a big gap compared to 94.65% of the standard training strategy, there is a nearly 1% improvement in the accuracy of the natural examples compared to the other three methods. (3) The robustness of both networks is significantly improved in the CAT training framework, which is higher than separately trained ones (51.02% vs 49.21% and 50.85% vs 48.48%). (4) The difference in accuracy between the two networks trained in the CAT framework is smaller than separately trained ones, which demonstrates that the two networks do well in collaborative learning. The robustness difference on CIFAR-10 between the two networks of TRADES-ALP against AA in the CAT training framework is 0.17%, while the difference is 0.73% under separate training. The same conclusion can be drawn from the results on CIFAR-100.

To investigate the generalizability of CAT, we conducted experiments with VGG-16 and MobileNet on CIFAR-10. The results are delayed to Appendices A.1 and A.2. Further, we conduct adversarial training with ResNet-18 on Tiny-ImageNet to explore the CAT on a large dataset, which is delayed to Appendix A.3. The robustness improvement holds true for all experiments.

### 4.2.2 BLACK-BOX ROBUSTNESS

For black-box attacks, we consider both transfer-based and query-based attacks. For the transfer-based attack, we use the standard adversarial training of ResNet-34 as the surrogate model, trained with the same parameters as described in Sec. 4. First, we perform the attack on the surrogate model to generate adversarial examples and then transfer them to the target network to get corresponding robustness. Here, we consider four attacks: FGSM, $PGD_{20}$, $PGD_{40}$, and $CW_\infty$, with the same attack parameters as Sec. 4.2.1. For query-based attacks, we consider the Square attack, which is more efficient. Tab. 3 shows the results, and CAT achieves the best performance. CAT can bring 1.79% and 3.19% robustness improvement against Square attack, for CIFAR-10 and CIFAR-100 respectively. Similarly, the improvement on CIFAR-100 is more significant than CIFAR-10.

## 4.3 COMPARISION TO SOTA

We use WideResNet-34-10 (Zagoruyko & Komodakis, 2016) networks for CAT to compare with previous sota methods. Tab. 4 shows the accuracy of the different methods for natural examples and the robustness against Auto-Attack. The robustness of both networks trained with CAT outper-

Table 4: **Quantitative comparison with previous methods.** WideResNet-34-10 is used in CAT. * denotes WideResNet-34-20, and † denotes WideResNet-40-8. AWP is equipped to get better robustness.

| Method | Clean | AA |
|---|---|---|
| Bag of Tricks for AT | 86.28 | 53.84 |
| HE* | 85.14 | 53.74 |
| Overfitting in AT* | 85.34 | 53.42 |
| Overfitting in AT | 85.18 | 53.14 |
| Self-Adaptive Training | 83.48 | 53.34 |
| FAT | 84.52 | 53.51 |
| TRADES | 84.92 | 53.08 |
| LLR† | 86.28 | 52.84 |
| LBGAT+TRADES ($\alpha = 0$)* | **88.70** | 53.57 |
| LBGAT+TRADES ($\alpha = 0$) | 88.22 | 52.86 |
| LBGAT+TRADES ($\alpha = 6$) | 81.98 | 53.14 |
| LAS-AT | 86.23 | 53.58 |
| LAS-AWP | 87.74 | 55.52 |
| CAT | 86.22 | 54.11 |
| | 86.51 | **54.20** |
| CAT+AWP | 86.74 | 56.43 |
| | 87.01 | **56.61** |

Table 5: **Quantitative comparison with KD-AT methods.** A WideResNet-34-10 and a ResNet-18 network are used in CAT to have a fair comparison with distillation methods. Time denotes training time (s) per epoch.

| Method | Stage | Time | Clean | AA |
|---|---|---|---|---|
| ARD | 2 | 2720 | 83.93 | 49.19 |
| IAD | 2 | 2723 | 83.24 | 49.10 |
| RSLAD | 2 | 2723 | 83.38 | 51.49 |
| RSLAD+AWP | 2 | - | 81.26 | 51.62 |
| CAT | 1 | 2516 | **84.39** | **51.72** |

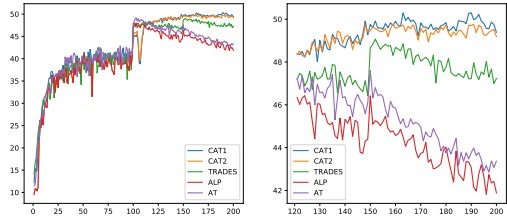

Table 6: **Robust accuracy of AT, ALP, TRADES, and CAT on CIFAR-10 during the adversarial training process.** CAT can alleviate the problem of overfitting.

forms the previous methods. Surprisingly, CAT with a smaller network even outperforms previous methods. AWP further boosts the robustness of CAT with a 2.41% improvement.

## 4.4 COMPARISION TO KD-AT

In general, the robustness of large models is higher than that of small models under the same training settings. For example, WideResNet-34-10 trained by TRADES can achieve 53.08% robustness against AA, while the robustness of ResNet-18 is only 49.21%. Researchers use knowledge distillation to distill the robustness of large models to small models and obtain remarkable results. We call these methods as KD-AT, which involves a teacher and a student network. Considering that CAT also involves two models, we compare CAT with previous KD-AT methods. To give a fair comparison, we use two different-size networks for CAT training, the same as the teacher and student network used in KD-AT. Note that, unlike the KD method where the teacher is trained in advance, CAT trains both the large model and the small model simultaneously. In other words, we extend previous offline distillation (2 stages) to an online way (1 stage) and achieve better performance with lower computation resources. Illustration comparison is shown in Appendix B.

Tab. 5 shows the results of KD-AT and CAT, where ARD (Goldblum et al., 2020), IAD (Zhu et al., 2021), and RSLAD (Zi et al., 2021) use WideResNet-34-10 trained by TRADES as teacher. CAT is collaboratively trained by two networks of different sizes. It can be seen that our method obtains high adversarial robustness and also obtains high clean accuracy. Meanwhile, CAT is more efficient than previous KD-AT methods, as shown in the 3-rd column. More importantly, The robustness of CAT is higher than RSLAD equipped with AWP.

## 5 PROPERTY ANALYSIS

### 5.1 ALLEVIATE OVERFITTING

Overfitting in adversarial training is first proposed by Rice et al. (2020), which shows the test robustness decreases after peak robustness. Overfitting is one of the most concerning problems in

Table 7: **The correlation between white-box robustness after CAT and prediction discrepancy (PD)** of different methods on CIFAR-10. ResNet-18 networks are used.

| Method | $\text{PGD}_{20}$ | PD |
|---|---|---|
| $\text{CAT}_{AT-ALP}$ | 53.11 | 18.98% |
| $\text{CAT}_{TRADES-ALP}$ | 54.44 | 21.05% |
| $\text{CAT}_{AT-TRADES}$ | 54.52 | 22.54% |

Table 8: **The white-box robustness of CAT with three networks on CIFAR-10**. Resnet-18 networks are used. T-A is short for TRADES-ALP. A-A-T is short for AT-ALP-TRADES

| Method | Clean | FGSM | $\text{PGD}_{20}$ | $\text{CW}_\infty$ | AA |
|---|---|---|---|---|---|
| $\text{CAT}_{T-A}$ | **84.75** | 59.76 | 54.17 | 52.72 | 50.85 |
| $\text{CAT}_{A-A-T}$ | 84.50 | 60.17 | 54.64 | 52.98 | 51.28 |
| | 84.62 | **60.25** | 54.87 | 53.04 | 51.42 |
| | 84.29 | 60.24 | **55.04** | **53.38** | **51.74** |

adversarial training. Here, we investigated the overfitting problem in CAT with VGG-16. Results are illustrated in Tab. 6. CAT can alleviate the overfitting problem that widely occurs in previous adversarial methods. Moreover, the performance for CAT has not saturated, and high performance is expected with longer epoch training.

## 5.2 CORRELATION OF DISCREPANCY AND CAT

To deepen the understanding of our CAT, we analyze the correlation between the discrepancy of different adversarial training methods and their adversarial robustness after CAT. First, we compute the prediction intersection between different methods, formulated as:

$$intersection = \frac{1}{N} \sum_{x_i \in D} \mathbb{I}(f(x_i), g(x_i)), \tag{7}$$

where $D$ is the datasets, and $\mathbb{I}$ is an indicator function, which is 1 when $f(x_i) = g(x_i)$ and 0 otherwise. Prediction discrepancy equals 1 minus intersection. The larger this value is, the greater the discrepancy. Then, we report the adversarial robustness of CAT trained in different settings. Results are reported in Tab. 7. A conclusion can be drawn that the greater the discrepancy between different methods is, the higher the adversarial robustness after CAT.

## 5.3 CAT OF THREE MODELS WITH THREE METHODS

CAT is a generalized method, which can use any number of different adversarial training methods for collaborative learning. To show the generalizability of CAT, we conducted an experiment on CAT by collaborating on three adversarial training methods. The results are reported in Tab. 8. The robustness improvement is more significant than CAT trained with two adversarial-trained methods. Collaborating three methods can bring 0.9% improvement against Auto-Attack, which shows the generalizability of CAT.

## 6 CONCLUSION

In this paper, we first analyze the properties of different adversarial training methods and find that networks trained by different methods perform differently on sample instances, *i.e.*, the network can correctly classify examples that are misclassified by other networks. Based on this observation, we propose a collaborative adversarial training framework to improve the robustness of both networks. CAT aims to guide network learning using true label supervision together with the knowledge mastered in peer networks, which is different from its own knowledge. Extensive experiments on different datasets and networks demonstrate the effectiveness of CAT. Furthermore, property analysis is conducted to get a better understanding of CAT. Broadly, CAT can be easily extended to multiple networks for collaborative adversarial training. We hope that CAT brings a new perspective to the study of adversarial training.

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

| Method | Clean | FGSM | PGD$_{20}$ | CW$_\infty$ | AA |
|--------|-------|------|-----------|------------|-----|
| AT | 78.31 | 53.11 | 48.39 | 46.32 | 43.69 |
| TRADES | 79.11 | 53.75 | 48.28 | 45.93 | 44.63 |
| ALP | 80.23 | 52.18 | 47.30 | 45.23 | 43.68 |
| CAT | 79.23 | 54.47 | **49.43** | 47.19 | **45.48** |
| | 80.12 | **54.48** | 48.30 | **47.23** | 45.33 |

| Method | Clean | FGSM | PGD$_{20}$ | CW$_\infty$ | AA |
|--------|-------|------|-----------|------------|-----|
| AT | 76.24 | 50.27 | 44.99 | 43.03 | 40.10 |
| TRADES | 75.84 | 49.65 | 45.26 | 42.04 | 41.08 |
| ALP | 79.46 | 50.14 | 43.95 | 42.08 | 40.01 |
| CAT | 80.14 | 51.25 | **46.38** | **44.24** | **42.20** |
| | 79.86 | **51.28** | 46.22 | 44.05 | 42.16 |

Table 9: The white-box robustness results (accuracy (%)) of CAT on CIFAR-10. We report the results of the best checkpoint. **VGG-16** networks are used in CAT framework.

Table 10: The white-box robustness results (accuracy (%)) of CAT on CIFAR-10. We report the results of the best checkpoint. **MobileNet** networks are used in CAT framework.

| Method | Clean | PGD$_{50}$ | CW$_\infty$ | AA |
|--------|-------|-----------|------------|-----|
| AT | 43.98 | 19.98 | 17.60 | 13.78 |
| TRADES | 39.16 | 15.74 | 12.92 | 12.32 |
| ALP | 39.85 | 17.28 | 15.34 | 12.98 |
| CAT | 44.35 | 20.86 | 19.43 | 14.96 |
| | **44.76** | **21.02** | **19.64** | **15.63** |

Table 11: The white-box robustness results (accuracy (%)) of CAT on **Tiny-ImageNet**. We report the results of the best checkpoint. Two ResNet-18 networks are used in CAT framework.

# A    MORE EXPERIMENTAL RESULTS

## A.1    VGG-16 RESULTS ON CIFAR-10

The white-box robustness of VGG-16 (Simonyan & Zisserman, 2014) models trained by AT, ALP, TRADES, and CAT are reported in Tab. 9. The setting for VGG-16 is the same as ResNet-18 models, i.e., $\alpha = 1.0/20$ and $\beta = 1.0/20$. The improvement for CAT with VGG-16 models is as consistent with ResNet-18 models. CAT can boost model's robustness under AutoAttack with 2.0 points.

## A.2    MOBILENET RESULTS ON CIFAR-10

Similar to the above VGG-16 models, we report the while-box robustness of MobileNet (Howard et al., 2017) on CIFAR-10 datasets under various attacks in Tab. 10. The experiment set is the same as the previous setting. We can see that CAT brings 1.0 improvement for MobileNet under AutoAttack, which is the most powerful adversarial attack method.

## A.3    RESNET-18 RESULTS ON TINY-IMAGENET

For the large-scale ImageNet dataset, just as all the baseline methods did not report the results, we are also unable to evaluate on ImageNet due to the very high training cost. To investigate the performance of CAT in large datasets, we conduct the experiment of white-box robustness of ResNet-18 on Tiny-ImageNet, which also is a widely used dataset in adversarial training. The results are shown in Tab. 11. Surprisingly, CAT shows impressive robustness on the large-scale dataset. The improvement is as significant as ResNet-18 in small datasets like CIFAR-10 and CIFAR-100.

## A.4    CAT OF ONE MODEL WITH VARIOUS ATTACKS

For CAT, we use two networks and two different attack methods for each network to perform adversarial training. An interesting baseline is one network with two different attack methods. Therefore, we use PGD and CW as our attack methods and one ResNet-18 as our network. The results are

| Method | Best Checkpoint | | | | |
| --- | --- | --- | --- | --- | --- |
| | Clean | FGSM | PGD$_{20}$ | CW$_\infty$ | AA |
| AT | 82.82 | 57.57 | 51.76 | 50.05 | 47.55 |
| CAT$_{T-A}$ | 83.91 | **59.76** | **54.44** | 52.56 | **51.02** |
| | **84.75** | **59.76** | 54.17 | **52.72** | 50.85 |
| CAT$_{P-C}$ | 82.09 | 56.48 | 52.48 | 49.28 | 48.06 |
| CAT$_{T-T}$ | 81.94 | 58.85 | 54.19 | 51.52 | 50.30 |
| | 82.13 | 58.77 | 54.02 | 51.56 | 50.14 |

Table 12: The white-box robustness results (accuracy (%)) of CAT on CIFAR-10. We report the results of the best checkpoint. P-C denotes one network trained by PGD and CW. T-A is short for TRADES-ALP, denoting two networks with TRADES and ALP. T-T is short for TRADES-TRADES, denoting two networks with TRADES and TRADES.

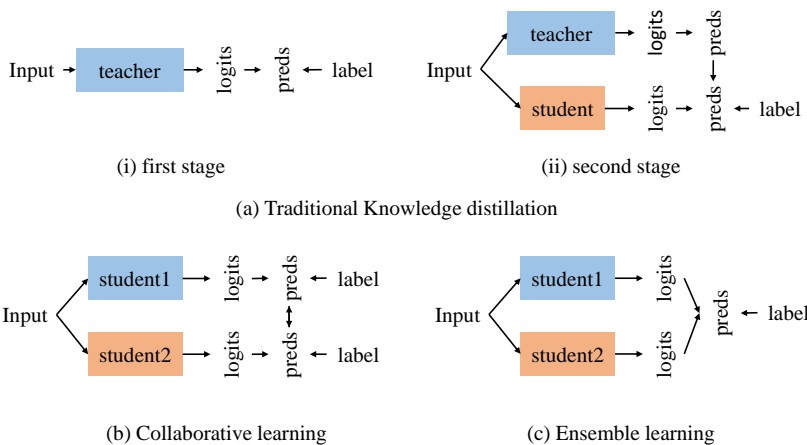

(i) first stage    (ii) second stage

(a) Traditional Knowledge distillation

(b) Collaborative learning    (c) Ensemble learning

Figure 4: Three types of distillation. (a) displays traditional knowledge distillation, which involves two-stage optimization and a large-scale teacher model. (b) and (c) illustrate online learning, *i.e.*, collaborative learning and ensemble learning, which do not involve teacher models.

reported in Tab. 12 (CAT$_{P-C}$ entry). The improvement for this setting is not significant as the previous setting, but it still, boosts the model's robustness against all four attacks.

### A.5    CAT OF TWO MODELS WITH SAME METHODS

Another interesting baseline is two networks trained by the same adversarial training methods, i.e., two ResNet-18 networks are both trained by TRADES. We denote this setting as CAT$_{T-T}$. The results are reported in Tab. 12. The improvement for this setting is not significant as the previous setting, but it still, boosts the model's robustness against all four attacks. However, the improvement is more significant than just using one network. A conclusion can be drawn that two networks are important for CAT to achieve better adversarial robustness.

## B    DISCUSSION

In this section, we illustrate three types of distillation methods, shown in Fig. 4. Traditional knowledge distillation has a two-stage optimization, which is pre-training the large-scale teacher model and distilling students with pre-trained teachers in the first stage. RSLAD (Zi et al., 2021) is implemented in this paradigm. Two-stage optimization brings a large computation cost. Compared to

RSLAD, CAT is based on collaborative learning and only needs one-stage optimization with two student models.

