# OpenReview forum: "CAT: Collaborative Adversarial Training"
_ICLR.cc/2024/Conference — ICLR 2024 Conference Withdrawn Submission_

### Official Review · Reviewer_aZir · 2023-10-14

**Soundness:** 3 good
**Presentation:** 2 fair
**Contribution:** 2 fair
**Rating:** 3
**Confidence:** 4

**Summary:**

This paper observes that different adversarial training methods exhibit distinct levels of robustness for sample instances. Based on this observation, the authors propose a collaborative adversarial training framework to enhance the robustness of neural networks by considering different kind of adversarial attacks.

**Strengths:**

- The performance of the proposed method looks good.

**Weaknesses:**

- The proposed method is trivial and not novel, it is just a modified version of TRADES.
- Lack of explanations about why such a modification of TRADES can lead to an improvement in performance.
- Lack of comparisons to the ensemble methods since ensemble methods are very relevant to the proposed method.
- The improvement seems negligible compared with AWP. The paper does not provide the previous SOTA method TRADES+AWP, which is reported to achieve an adversarial accuracy of $56.17\\%$ (under AutoAttack) in the original paper of AWP. Compared with TRADES, CAT improves the adversarial accuracy to $54.2\\%$, while AWP improves to $56.17\\%$, which is much better than CAT. Given the existence of AWP, adding CAT only improves the adversarial accuracy to $56.61\\%$, where the improvement is no more than $0.5\\%$, so I think there is no necessity to combine CAT with AWP, we can use AWP, since CAT leads to additional computations.
- The paper reports the best of the two in CAT models as the performance of CAT, however, in practice, how do you choose which model to use? The paper lacks discussions on this point.

**Questions:**

- See the weaknesses.
- The current SOTA method uses additional data generated by diffusion models to get a more robust model, it is not known that whether the improvement of CAT still exists when we use a lot of additional generated data. Perhaps CAT shows no improvement in this case, just as the case when we use AWP, the improvement of CAT is no more than $0.5\\%$.

---

### Official Review · Reviewer_9NK1 · 2023-10-31

**Soundness:** 2 fair
**Presentation:** 3 good
**Contribution:** 2 fair
**Rating:** 5
**Confidence:** 2

**Summary:**

- Draft presents an observation that different adversarial training methods, despite being very close in numerical comparison of their adversarial accuracy, have a significant prediction discrepancy. Harnessing that observation, it presents a collaboration framework for fusing the desired behavior from different adversarial training methods.
- The proposed method, Collaborative Adversarial Training (CAT), proposes to utilize the (KL divergence of) logits result of multiple adversarial training to exert the desired collaboration.
- Experiments (in the main draft) are performed on CIFAR-10 and CIFAR-100. Experiments demonstrate that the proposed framework can fuse multiple adversarial training objectives to collaborate. Results show slight performance improvement as a result of the CAT.
- Other ablations and analysis discusses the hyperparameters and behavior of the proposed method (overfitting, correlation with the prediction discrepancy, etc.)

**Strengths:**

+ The proposed method is very intuitive. It aims to adopt the best of multiple worlds.
+ The presentation is very clear and easy to understand.

**Weaknesses:**

1. One of the main drawbacks of the proposed approach is that it fails to convince that the collaboration always picks the best from the participating adversarial training methods. This is a general issue in any fusion-based approach. Given that the individual adversarial training methods have nontrivial prediction discrepancy, how does the proposed framework ensure that the collaboration always picks the best from individual methods?
2. From Figure $3$, it is clear that the proposed CAT framework is not very effective. For instance, for the model trained using TRADES, the gain in the adversarial accuracy (across $3$ different adversaries) is not more than $1$\%. Notably, the peak performance occurs at a relatively lower value of $\alpha$. Furthermore, the performance appears to degrade swiftly. Observing the ablation results beyond $\alpha=0.2$ would have been complete. Similarly, for the model trained using AT (red curve), the maximum performance gain from the proposed collaboration is about $1$\% (that too against the simplest of adversarial attacks, FGSM).
3. This observation can also be derived from Tables $1$ and $2$. The maximum gain from the proposed collaboration against the strongest adversarial attack (AA; best checkpoint) is not even $3$\%.
4. Experiments to compare against the SOTA (Table $4$) also reveal a similar picture. Improvement of the proposed CAT over the AT+tricks is less than $.5$\%; AWP improves the performance of CAT, that too only about $2$\%.
5. Since the main contribution of the draft is strongly intuitive, it needs to be supported by strong experiments. However, the experimental evaluation presented in the draft considers relatively simpler datasets (CIFAR). The object recognition models trained on the ILSVRC would have strengthened the experimental evaluation.

**Questions:**

- Please refer to the weaknesses section.
Minor
- Figure 1 is not very clear. What is captured by each cell in the confusion matrix? Generally, we see a strong diagonal presence in confusion matrices. However, in this case, the diagonal is zero, and the off-diagonal elements appear to dominate. Authors may explain this.
- Equation 1 needs to be corrected. The 'argmax' needs to be replaced by 'max.'

---

### Official Review · Reviewer_VPkg · 2023-10-31

**Soundness:** 3 good
**Presentation:** 2 fair
**Contribution:** 3 good
**Rating:** 5
**Confidence:** 4

**Summary:**

The authors propose an adversarial defense method called collaborative adversarial training, which combines multiple adversarial training algorithms together by feeding their generated adversarial examples into each other. Experimental evaluations show the method improves upon state-of-art in defending against strong attacks like AutoAttack.

**Strengths:**

- The proposed method works well empirically, achieving improvements in robust accuracy against white box attacks by AutoAttack compared to other state-of-art methods.

- The idea of using adversarial examples generated by different models to improve robustness makes sense, and proves to be quite useful.

**Weaknesses:**

- I am conerned with the writing clarity of the paper. There are numerous issues with in Section 3.
  1. The argmax in Eqt 1 should be max instead, because the argmax is an input image, not a scalar loss value. The notations for x^adv_AT, x^adv_TRADES is also not particularly rigorous as it does not make the dependence on the underlying functions f_\theta, g_\theta explicit.
  2. Similarly, without a set of clear notations, Eqt 6 is very confusing. Which version of L_TRADES are we referring to here? For f or for g?
  3. Also, in 3.2 f is used to denote the network trained by AT. But if we add adversarial examples generated by g into the training of f, then f is no longer than the same as the f that we would obtain by vanilla AT. So what exactly are we trying to do in Figure 2? Without precise notations it is very difficult to understand or reproduce the method properly.

- What is ALP and AWP referred to in the paper? They are used directly without a reference or the non-abbreviated form. If ALP is adversarial logit pairing, there have been doubts on whether the model is truly robust (see paper below). Then why are the CAT models trained with ALP performing so well?

Evaluating and Understanding the Robustness of
Adversarial Logit Pairing
Engstrom et al
https://arxiv.org/pdf/1807.10272.pdf

**Questions:**

- In Section 4.2.1 first sentence, it seems to be implied that AA is run with 20 iterations. But that is not the default setting for AA (should be at least 100). So what is the number of iterations used in AA, as this could affect comparison with other papers?

- The authors need to improve the clarity of the algorithm description to improve the score for the paper.

---

### Official Review · Reviewer_V2Su · 2023-11-01

**Soundness:** 2 fair
**Presentation:** 3 good
**Contribution:** 3 good
**Rating:** 5
**Confidence:** 4

**Summary:**

The paper highlights the discrepancy in the outputs of models trained using different adversarial training methods. Based on this observation, a novel adversarial training method named Collaborative Adversarial Training is proposed. The proposed method simultaneously trains multiple models using different AT methods. During training, the  adversarial samples generated by one network are used by other networks to facilitate knowledge exchange. The effectiveness of the proposed approach is demonstrated by considering various networks  (ResNet, VGG, and MobileNet) and datasets (CIFAR and Tiny-ImageNet).

**Strengths:**

The paper presents an intriguing observation on models trained using various AT methods (i.e., the models' prediction discrepancy). To address this prediction discrepancy, the paper proposes a simple and generic solution. The proposed AT method yields models with a notable improvement in robustness (except for the CIFAR-10 dataset).

**Weaknesses:**

1. The paper presents an important observation, but it fails to explain the cause for this prediction discrepancy. Further, it fails to explain why prediction discrepancy affects adversarial robustness. A thorough analysis of the observation is missing, such as (a) prediction discrepancy for models obtained via different runs for a given training method, and (b) analysis at the category level.
2. For CIFAR-10, the improvement in the robustness is not significant when compared with TRADES (table-2) and TRADES-AWP [1] (comparison is missing in table-4).
3. Minor: Mention the values of hyperparameters of TRADES and ALP. Provide references for the methods compared in Table 4.

[1] Wu, Dongxian, Shu-Tao Xia, and Yisen Wang. "Adversarial weight perturbation helps robust generalization." Advances in Neural Information Processing Systems 33 (2020): 2958-2969.

**Questions:**

Address weaknesses.